# LEO: A Graph Attention-Based Framework for Learned Object Extensions and Adaptive Sensor Fusion for Autonomous Driving Applications

## Abstract

Accurate shape and trajectory estimation of dynamic objects is a fundamental requirement for reliable perception in Automated Driving (AD). In the classical versions of AD algorithms and stacks, various Bayesian extended object geometric models are used to provide object-related extensions and trajectories. Performance of such approaches are deeply connected with the completeness of a-priori and update-likelihood functions. Recent deep learning approaches improve flexibility by learning shape features directly from raw or fused sensor data, but they often rely on dense annotated datasets and high computational resources, which restricts their applicability in production vehicles. We aim to improve production-level automated driving systems by integrating the computational efficiency and theoretical robustness of geometric methods with the adaptability and generalization capabilities of modern deep learning techniques. We employ a task-specific parallelogram-based ground-truth formulation to represent object extensions, facilitating expressive modeling of complex geometries such as articulated trucks and trailers. Our primary contribution is the development of a novel spatio-temporal Graph Attention Network (GAT)-based model, Learned Extension of Objects (LEO), that demonstrates proficiency in adaptive fusion weight learning, temporal consistency, and multi-scale shape representation from multi-modal production grade sensor tracks. LEO successfully generalizes across various sensor modalities, configurations, object classes, and geographic regions, exhibiting robustness even under challenging conditions and longer range targets. We have presented these observations and evaluations based on the real-world Mercedes-Benz SAE Level-3 (L3) DRIVE PILOT dataset in our article. Furthermore, its computational efficiency makes it a suitable candidate for integration into a real-time production system, although further validation and integration efforts are necessary for deployment in safety-critical systems.

## 1 Introduction

AD has emerged as a transformative paradigm for improving road safety, mobility, and efficiency in modern transportation. Human error accounts for nearly 94% of severe accidents, highlighting the potential of Autonomous Vehicles (AVs) to enhance safety through consistent, rule-based decision making and improved situational awareness (Singh, 2015). Beyond safety, AD promises extended mobility for elderly and disabled users, reduced congestion via coordinated routing, and lower costs through fuel efficiency and shared ownership models (Fagnant & Kockelman, 2015; Yurtsever et al., 2020). These advantages have fueled substantial research and industrial investment, positioning AD as a cornerstone of future intelligent transportation systems (Badue et al., 2021).

The deployment of AVs relies on the integration of perception, prediction, planning, and control, with perception forming the foundation (Li & Ibanez-Guzman, 2020). Multi-modal sensor suites integrating LiDAR, RADAR, and cameras are commonly employed in contemporary systems to leverage their respective strengths. LiDAR provides high-resolution geometric data, albeit with diminished point cloud density at extended ranges. RADAR offers robust velocity measurements and

resilience to adverse environmental conditions, notwithstanding its limited spatial resolution. Cameras furnish rich semantic information, but lack inherent precise depth perception (Yeong et al., 2021). Robust sensor fusion is thus essential for holistic scene understanding and safe decision making (Arnold et al., 2019). A key challenge in perception is accurate estimation of object geometry. Many tracking methods simplify targets as points, neglecting spatial extent. In real traffic, however, vehicles, cyclists, and pedestrians occupy significant space and typically generate multiple measurements per frame. This motivates Extended Object Tracking (EOT), which jointly estimates kinematics and shape (Koch, 2016). Reliable shape estimation is particularly critical in dense urban scenarios with vulnerable road users, where inaccurate modeling can lead to unsafe distance keeping or unnecessary evasive maneuvers.

Classical EOT approaches, such as random matrix models (Feldmann et al., 2010; Haag et al., 2018), provide efficient ellipse approximations but degrade under occlusions and articulated shapes. Non-parametric contour formulations, including Gaussian processes (Granstrom et al., 2016), improve geometric flexibility but rely on dense observations and incur high computational costs. More recently, learning-based methods estimate shape features directly from raw or fused sensor data (Meyer & Thakurdesai, 2020; Dong et al., 2020), alleviating parametric limitations yet facing challenges with annotation costs, generalization across sensor configurations, and robustness under sparse or noisy conditions (Wang et al., 2021). In this context, Graph Neural Networks (GNNs) have emerged as a powerful paradigm for modeling spatial relationships and temporal dependencies in structured automotive perception data (Wang et al., 2019), including learned-geometry approaches such as the Graph Transformer in 3DMOTFormer (Ding et al., 2023). While curated datasets such as KITTI (Geiger et al., 2013), nuScenes (Caesar et al., 2020), and Waymo (Sun et al., 2020a) have enabled the development of these increasingly complex models, production systems must operate under stringent computational and bandwidth constraints, often exposing only object-level tracks rather than raw sensor measurements (Duraisamy et al., 2013). These restrictions limit the applicability of dense point-cloud architectures and motivate the need for data- and compute-efficient formulations.

To address these challenges, this work introduces the **L**earned **E**xtension of **O**bjects (LEO) framework for production-oriented extended object tracking. The key contributions are:

- A spatio-temporal architecture that leverages Graph Attention Network (GAT) blocks, originally proposed by Veličković et al. (2018), to enable adaptive shape estimation under production constraints.

- A parallelogram-based ground-truth formulation that generalizes bounding geometries to represent both rectangular and articulated objects such as trucks with trailers.

- A dual-attention mechanism that jointly captures intra-modal temporal dynamics and inter-modal spatial dependencies across multi-sensor tracks for robust fusion and sequential learning.

- Comprehensive evaluation on large-scale, real-world automotive datasets, demonstrating accurate, and computationally efficient performance across diverse driving scenarios.

## 2 RELATED WORKS

**Deep Learning for Object Detection**   The advent of deep learning has enabled models to learn complex geometric representations directly from multi-modal datasets with ground-truth 3D annotations. Early CNN-based approaches, such as PointPillars and SECOND (Lang et al., 2019; Yan et al., 2018), process voxelized inputs to produce oriented bounding boxes efficiently, while point-based methods like PointNet++ (Qi et al., 2017) operate directly on raw point clouds. Transformer-based architectures, including DETR3D and BEVFormer (Wang et al., 2022; Li et al., 2024), exploit attention in Bird's-Eye View representations. Multi-modal fusion strategies, e.g., camera-LiDAR-RADAR integration (Yeong et al., 2021; Bai et al., 2022), further enhance robustness under challenging conditions. Recent end-to-end EOT frameworks, such as CenterTrack (Zhou et al., 2020), TrackFormer (Meinhardt et al., 2022), and TransTrack (Sun et al., 2020b), integrate detection, association, and shape estimation in a unified pipeline. By leveraging temporal embeddings and attention mechanisms, these models maintain object identities and consistent shape estimates across frames, even under occlusions or missed detections.

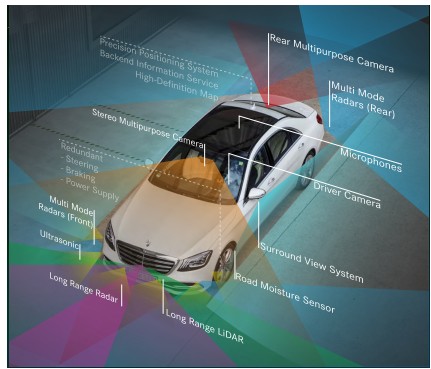

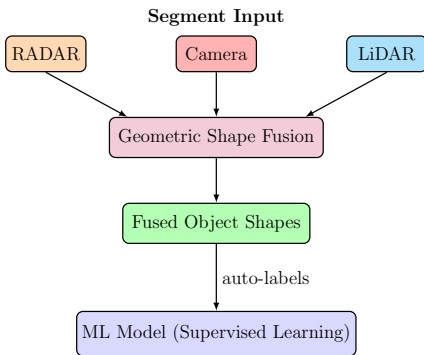

(a) Mercedes-Benz EQS sensors.

(b) Supervised Learning using labels from geometric method

Figure 1: Mercedes-Benz EQS sensors used by DRIVE PILOT Mercedes-Benz (2023) (a) and auto-labelling pipeline (b).

**Extended Object Tracking** Classical Extended Object Tracking (EOT) used Bayesian filters with simple parametric shapes like ellipses (Feldmann et al., 2010; Lan & Li, 2019), offering efficiency but limited expressiveness. Learning-based tracking integrates temporal consistency via transformers, exemplified by TrackFormer and TransTrack (Meinhardt et al., 2022; Sun et al., 2020b). However, there is not much literature on deep-learned extension of objects in the context of EOT.

## 3 GEOMETRIC METHOD AND AUTO-LABELING

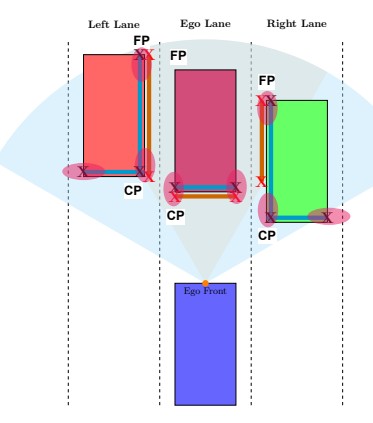

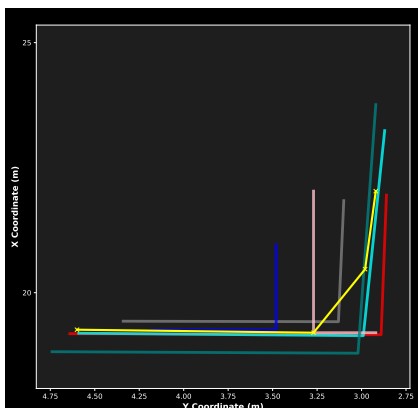

(a) Sensor target shape types

(b) Target as detected by multiple sensors

Figure 2: Comparison of sensor field-of-view–based shape abstractions under occlusions, with overlays of the FOVs for RADAR (60°) and LiDAR (120°), along with oval-shaped extension point covariances and lane-wise points for evaluation (a) and multi-sensor target shape segments (b).

In series-production vehicles, raw sensor data is typically unavailable due to bandwidth, certification, and proprietary constraints from suppliers, resulting in perception modules that output high-level object tracks rather than low-level measurements (Duraisamy et al., 2013). These sensor tracks contain kinematic estimates, classification attributes, state covariances, and coarse object extents, abstracting away raw point clouds or image detections. (Duraisamy et al., 2015) presents combination of this information granularity (Duraisamy et al., 2023) to achieve improved data association and fusion quality. Track-level fusion has emerged as a practical paradigm Bar-Shalom et al. (2001); Tian et al. (2012), enabling modular integration of sensors and robustness across automotive platforms. Each sensor delivers object hypotheses in the form

$$List_{\text{sens}} = \{\hat{\mathbf{x}}_i, \mathbf{P}_i, Ext_j\} \tag{1}$$

where $\hat{\mathbf{x}}_i$ is the estimated kinematic state, $\mathbf{P}_i$ the covariance, and $Ext_j$ the $j$-th extension point with $m \leq 3$ depending on sensor resolution. The fusion task defines a function that generates a consistent fused representation of objects in Equation 2.

$$FusedShape = f(\hat{x}_{sens,i}, P_{sens,i}) \tag{2}$$

Objects are abstracted as primitive geometric types depending on sensor modality and resolution depicted in Figure 2a: **L-shapes** for high-resolution sensors like LiDAR capturing both edges and object in sensor's FOV, **I-shapes** when only one edge is visible, such as vehicle in front of ego vehicle or occluded, and **point-shapes** typical of RADAR with limited resolution at far ranges. This representation enables handling heterogeneity and partial observability across modalities. The hybrid fusion framework is modular, comprising kinematic state fusion with Kalman Filter (KF) or Covariance Intersection (CI), and shape extension fusion using computational geometry (Duraisamy et al., 2016; 2023). Segment association relies on spatial and orientation criteria, using Hausdorff distance with threshold $d_{\text{Hausdorff}} < 2\,\text{m}$ and angular constraint $\theta < 30°$.

$$d_{Hausdorff} = \max\left(d(S_1, S_2), d(S_2, S_1)\right) \tag{3}$$

Once the association is established, segment endpoints are confidence-weighted inversely with their covariance determinant

$$Weight \propto \frac{1}{|\Sigma|} \tag{4}$$

prioritizing high-certainty observations. To conservatively combine correlated sensor data, Covariance Intersection (CI) is used, e.g.,

$$\sum_{FusionStart}^{-1} = \omega \sum_{S1start}^{-1} + (1 - \omega) \sum_{S2start}^{-1} \tag{5}$$

with $\omega \in [0, 1]$ balancing uncertainty contributions. Experimental validation on a Mercedes-Benz prototype with RADAR, LiDAR, and stereo cameras demonstrated sub-$10\,\text{cm}$ lateral accuracy, full modularity at the track level, and industrial readiness, highlighting the suitability of track-level fusion for safety-certified automotive perception stacks. In continuation of this model-based approach, the fused object shapes having three extension points serve as reliable auto-labels, fused tracks (Figure 4), that are subsequently utilized to supervise the training of LEO (Haag et al., 2020). As illustrated in Figure 1b, this establishes a closed-loop framework where geometric fusion not only enables modular perception in production systems but also provides consistent training targets for data-driven methods, thereby bridging model-based and learning-based paradigms within the automotive perception stack.

## 4 LEO: GRAPH ATTENTION NETWORK BASED SHAPE ESTIMATION

**Parallelogram-Based Object Representation** Traditional rectangular bounding boxes inadequately capture articulated or disjoint geometries, such as trucks with trailers. Since the sensor tracks in our dataset do not impose right-angle constraints, we represent objects as parallelograms, where the fourth vertex is obtained by completing the shape from three ordered extension points of the fused objects from geometric fusion. Each object is parameterized by its left rear vertex (reference point: $RF_x, RF_y$), dimensions $(l, w)$, orientation and internal angle $(\theta, \theta^*)$, and velocities $(v_x, v_y)$, following the DIN 70000 standard (Haken, 2015). This formulation generalizes rectangular cases $(\theta^* = 90°)$ while accommodating complex geometries through flexible angular constraints, as illustrated in Figure 3a. The resulting state vector or label is:

$$\hat{\mathbf{y}} = \{RF_x, RF_y, l, w, \theta, \theta^*, v_x, v_y\} \in \mathbb{R}^8 \tag{6}$$

### 4.1 PROBLEM FORMULATION AND GRAPH CONSTRUCTION

We formulate multi-modal sensor fusion as a spatio-temporal graph learning problem Fey & Lenssen (2019) over heterogeneous sensor measurements with varying sampling rates as illustrated in Figure 3b. The temporal alignment pipeline processes raw measurements from RADAR ($60\,\text{Hz}$), LiDAR ($40\,\text{Hz}$), and cameras ($80\,\text{Hz}$) through dedicated trackers, synchronizing outputs in $20\,\text{ms}$ intervals within a $120\,\text{ms}$ sliding window, producing target states and extension points (Figure 2b).

As sensors fire asynchronously at different frequencies, missing detections at a given timestamp are handled by propagating the most recent measurement in the data stream. Shape cues, primarily from LiDAR contours, are abstracted into L-shapes using geometric feature extraction and a dual-line RANSAC procedure (Ling et al., 2024) for robustness against outliers.

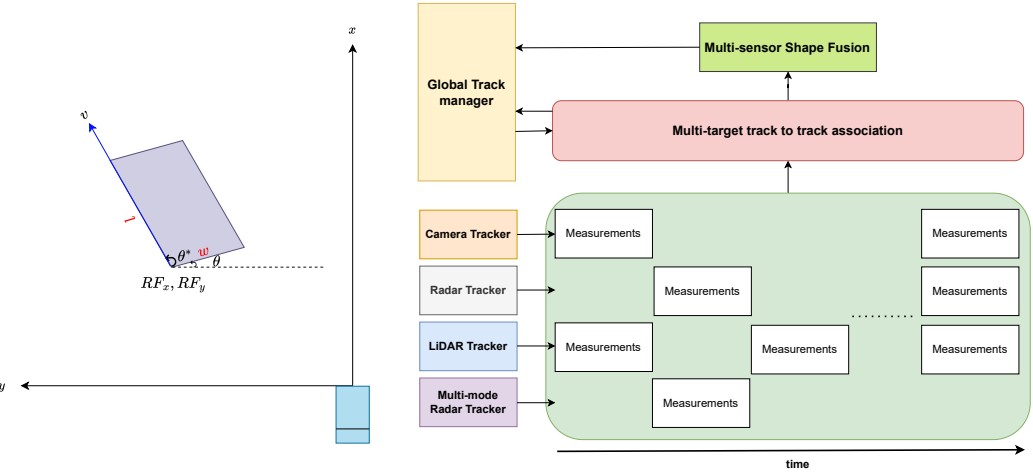

(a) Parallelogram-shaped object representation in the ego coordinate frame.

(b) Shape Fusion Architecture

Figure 3: Parallelogram object representation with velocity vector represented as an arrow from the reference point, which is at the left-rear vertex (a) and the proposed Shape Fusion architecture (b).

The spatio-temporal graph $\mathcal{G} = (\mathcal{V}, \mathcal{E})$ comprises 48 nodes: 6 ego-motion nodes encoding velocity, yaw rate, acceleration, and timestamp and 42 sensor nodes from seven modalities (**L**ong-**R**ange **Li**DAR, **L**ong-**R**ange **R**ADAR, **M**ulti-**M**ode **R**ADAR **F**ront **R**ight, **M**ulti-**M**ode **R**ADAR **F**ront **L**eft, **M**ulti-**P**urpose **C**amera, **Li**DAR contour, and **S**tereo **M**ulti-**P**urpose **C**amera) across six timestamps. Each sensor node $\mathbf{n}_{s,t-k}$ encodes an 11-dimensional feature vector:

$$\mathbf{f}^{(t-k)} = [x_1, x_2, x_3, y_1, y_2, y_3, \sigma_x^2, \sigma_y^2, v_x, v_y, \Delta t]^T \tag{7}$$

representing extension points $x_i, y_i$, uncertainties $\sigma_x, \sigma_y$, velocities $v_x, v_y$, and temporal offset $\Delta t$ in seconds to the fusion timestamp. Ego-motion nodes are similarly encoded as

$$\mathbf{n}_{\text{ego},t-k} = [v_{t-k}, \dot{\psi}_{t-k}, a_{t-k}, \ldots, \Delta t_k]^T \in \mathbb{R}^{11}, \tag{8}$$

allowing implicit learning of ego-motion compensation. The edge set $\mathcal{E}$ captures temporal evolution and cross-modal dependencies through three edge types:

$$\mathcal{E}_{\text{temporal}} = \{(\mathbf{n}_{s,t-k}, \mathbf{n}_{s,t-(k-1)}) \mid s \in [1,8], \, k \in [1,5]\} \tag{9}$$

$$\mathcal{E}_{\text{spatial}} = \{(\mathbf{n}_{s_i,t-k}, \mathbf{n}_{s_j,t-k}) \mid s_i \neq s_j, \, k \in [0,5]\} \tag{10}$$

$$\mathcal{E}_{\text{self}} = \{(\mathbf{n}_{s,t-k}, \mathbf{n}_{s,t-k}) \mid s \in [1,8], \, k \in [0,5]\} \tag{11}$$

### 4.2 DUAL ATTENTION MECHANISM, TRAINING AND NETWORK ARCHITECTURE

LEO employs a dual-attention mechanism (Figure 5) that independently models temporal consistency, i.e., shape evolution and motion dynamics, within individual sensor modalities (intra-modal), while simultaneously integrating complementary spatial information across modalities (inter-modal) (Veličković et al., 2018). The resulting unified attention formulation is given by:

$$\alpha_{ij}^{(m)} = \frac{\exp\left(\text{LeakyReLU}\left(\mathbf{a}_m^\top [\mathbf{W}_m \mathbf{h}_i \,\|\, \mathbf{W}_m \mathbf{h}_j]\right)\right)}{\sum_{k \in \mathcal{N}_i^{(m)}} \exp\left(\text{LeakyReLU}\left(\mathbf{a}_m^\top [\mathbf{W}_m \mathbf{h}_i \,\|\, \mathbf{W}_m \mathbf{h}_k]\right)\right)} \tag{12}$$

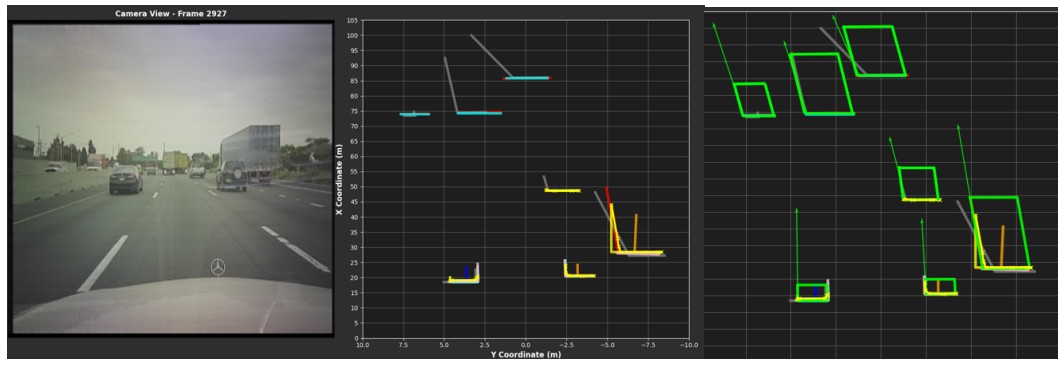

(a) Input multi-sensor tracks                                    (b) Fused tracks

Figure 4: Depiction of associated input multi-modal tracks to global tracks in a scene(a) and final fused tracks after doing geometric fusion which is used as labels for training LEO (b).

where $m$ denotes the attention type: intra corresponds to temporal neighbors $\mathcal{N}_i^{\text{temporal}}$, capturing motion-consistent patterns, while inter corresponds to spatial neighbors $\mathcal{N}_i^{\text{spatial}}$, aggregating complementary information across modalities. $\mathbf{W}_m$ and $\mathbf{a}_m$ are the learnable weight and attention query vectors for the respective modality.

The final attention coefficients balance temporal and spatial contributions:

$$\alpha_{ij}^{\text{st}} = \lambda \cdot \alpha_{ij}^{\text{intra}} + (1 - \lambda) \cdot \alpha_{ij}^{\text{inter}} \tag{13}$$

enabling adaptive weighting based on data availability and quality. Message passing follows:

$$\mathbf{h}_i^{(l+1)} = \sigma \left( \sum_{j \in \mathcal{N}_i} \alpha_{ij}^{\text{st}} \mathbf{W}^{(l)} \mathbf{h}_j^{(l)} \right) \tag{14}$$

**Training Objective and Optimization**    The training objective combines parameter-level regression with geometry-aware supervision through a composite loss function:

$$\mathcal{L}_{\text{total}} = \mathcal{L}_{\text{param}} + \lambda_{\text{IoU}} \mathcal{L}_{\text{IoU}} \tag{15}$$

The parameter loss applies SmoothL1 regression to individual components:

$$\mathcal{L}_{\text{param}} = \sum_{i \in \{RF_x, RF_y, l, w, \theta, \theta^*, v_x, v_y\}} \beta_i \cdot \text{SmoothL1}(\hat{\mathbf{y}}_i, \mathbf{y}_i) \tag{16}$$

where $\beta_i$ weights balance parameter importance based on estimation difficulty and downstream impact. The geometry loss combines Generalized IoU Rezatofighi et al. (2019) and Distance IoU Zheng et al. (2020) to enforce spatial consistency:

$$\mathcal{L}_{\text{IoU}} = \alpha \cdot \mathcal{L}_{\text{GIoU}} + (1 - \alpha) \cdot \mathcal{L}_{\text{DIoU}} \tag{17}$$

where GIoU ensures enclosure constraints while DIoU enforces centroid alignment. Training is conducted using the Adam optimizer (Diederik P. Kingma, 2015) with an initial learning rate of $1 \times 10^{-4}$ and plateau-based decay (factor 0.75). The loss function uses $\beta = 1$ and $\alpha = 0.5$. The model is trained for up to 50 epochs with a batch size of 128 and gradient clipping at a norm of 3.0. Early stopping with a patience of 5 epochs is applied to prevent overfitting, with convergence typically achieved around 40 epochs, beyond which validation performance stagnates.

## 5    EVALUATION

**Dataset Description**    The proposed model is evaluated on proprietary data collected from the Mercedes-Benz SAE Level-3 DRIVE PILOT system. The dataset comprises multi-sensor fusion

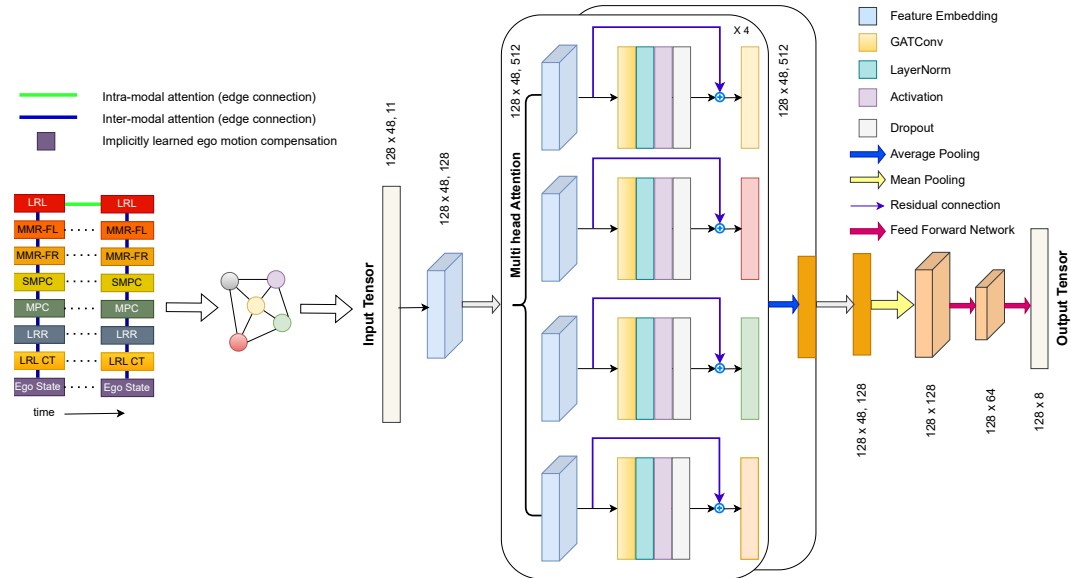

Figure 5: LEO architecture : Tracks from multi-modality sensors are first embedded with state vectors and timestamps, concatenated across six frames (120 ms), and represented as a spatio-temporal graph with intra- and inter-modal edges for GAT-based attention. The LEO architecture then projects inputs ($128 \times 48 \times 11$) into a latent space ($128 \times 48 \times 128$), processes them through four stacked GAT-Conv Veličković et al. (2018) layers with dual attention, normalization, ELU activation, dropout, and residual connections, and aggregates multi-head outputs via pooling into $128 \times 128$ embeddings. A final feed-forward projection maps these to $128 \times 8$ parallelogram parameters $\hat{\mathbf{y}}$, enabling efficient joint spatio-temporal reasoning for shape fusion.

outputs of static and dynamic objects, combined with ego vehicle states across a wide range of driving environments in the United States and Europe. It is partitioned into a training set of $12.3\,\mathrm{h}$ and a testing set of $2.31\,\mathrm{h}$, having a mix of highway driving and *cut-in* sequences (Table 1). The *cut-in* sequences originate from controlled proving ground experiments designed to enrich safety-critical coverage. It covers diverse traffic participants including passenger cars, commercial vehicles, articulated trucks, and vulnerable road users. Sensor fusion provides balanced multi-lane coverage over all objects within $RF_x \in [-10, 100]\,\mathrm{m}$ and $RF_y \in [-12, 12]\,\mathrm{m}$ (ROI), with object dimensions ranging from compact cars ($\approx 3\,\mathrm{m}$) to articulated vehicles exceeding $70\,\mathrm{m}$. Ego states span urban to highway conditions with velocities up to $140\,\mathrm{km/h}$, yaw rates within $\pm 0.6\,\mathrm{rad/s}$, and accelerations between $-10$ and $+5\,\mathrm{m/s^2}$. The velocity data highlights this variability, showing dominant longitudinal motion alongside critical lateral maneuvers such as cut-ins, overtakes, and lane changes. This diversity ensures that both common traffic flow and safety-critical events are well represented, establishing a production-relevant benchmark for evaluation.

Table 1: Dataset composition for training and testing sequences. "Cut-Ins" correspond to proving ground data emphasizing safety-critical maneuvers.

|  | Driving | Cut-Ins | Hours | Fusion Objects |
|---|---|---|---|---|
| Train Sequence | 326 | 410 | 12.3 hrs | 1.46 mil. |
| Test Sequence | 79 | 60 | 2.31 hrs | 0.44 mil. |

## 5.1 EVALUATION STRATEGY

Evaluation is conducted on the complete test dataset, using region-based overlaps of oriented parallelograms (GIoU and DIoU) and the Mean Absolute Error (MAE) of output parameters. Objects are stratified by length, with $l_1 \in [3, 10]\,\mathrm{m}$ representing cars and light commercial vans, and $l_2 > 10\,\mathrm{m}$ representing buses, trucks and trailers. The evaluation is reported along two complementary axes:

first, *global performance* across all objects in the ROI, providing an overall benchmark of model robustness; and second, a *lane-wise analysis*, where results are partitioned by object centroid position into ego lane (EL: $[-1.5, 1.5]$ m), left lane (LL: $(1.5, 4.5]$ m), and right lane (RL: $[-4.5, -1.5)$ m), as depicted in Figure 2a. This structure ensures that both aggregate accuracy and spatially resolved safety-critical contexts for motion planning are systematically assessed.

Table 2: Global KPIs for Shape Estimation of Fused Objects for LEO

| Parameter | $l_1$ | | | | $l_2$ | | | |
| | with $\alpha_{ij}^{\text{inter}}$ | | without $\alpha_{ij}^{\text{inter}}$ | | with $\alpha_{ij}^{\text{inter}}$ | | without $\alpha_{ij}^{\text{inter}}$ | |
| | MAE | Error (%) | MAE | Error (%) | MAE | Error (%) | MAE | Error (%) |
|---|---|---|---|---|---|---|---|---|
| GIoU (-) | 0.78 | – | 0.27 | – | 0.76 | – | 0.31 | – |
| DIoU (-) | 0.82 | – | 0.23 | – | 0.76 | – | 0.34 | – |
| $RF_x$ (m) | 0.21 | 0.60 | 3.98 | 11.45 | 0.40 | 1.35 | 3.18 | 10.49 |
| $RF_y$ (m) | 0.11 | 2.94 | 0.64 | 16.42 | 0.14 | 4.95 | 0.67 | 23.67 |
| $l$ (m) | 0.43 | 10.16 | 3.55 | 83.29 | 2.22 | 11.62 | 8.60 | 44.88 |
| $w$ (m) | 0.08 | 4.88 | 0.37 | 21.03 | 0.12 | 5.22 | 0.27 | 11.30 |
| $\theta$ (rad) | 0.04 | – | 0.13 | – | 0.05 | – | 0.11 | – |
| $\theta^*$ (rad) | 0.05 | 3.09 | 0.13 | 8.13 | 0.05 | 3.24 | 0.11 | 6.98 |
| $v_x$ (m/s) | 0.24 | 2.01 | 3.37 | 28.15 | 0.30 | 3.27 | 2.73 | 29.07 |
| $v_y$ (m/s) | 0.10 | – | 0.23 | – | 0.12 | – | 0.21 | – |

**Global Performance**   LEO achieves high spatial accuracy with GIoU/DIoU scores of $0.76$–$0.82$ across both object categories. Reference point estimation remains below $0.4$ m (MAE) with relative errors under $5\%$, while dimensional accuracy is consistent: car-sized objects ($l_1$) attain MAE of $0.43$ m in length and $0.08$ m in width, and articulated objects ($l_2$) reach $2.22$ m and $0.12$ m, corresponding to $10$–$12\%$ relative errors. Orientation errors remain below $3°$ and velocity estimates are precise within $0.3$ m/s ($< 1.3$ km/h). Implemented in PyTorch and benchmarked on an RTX 2080 Ti GPU with an 18-core CPU, LEO processes samples at avg. inference time $\sim 13.5$ ms (runtime $\sim 30$ FPS) with minimal memory usage ($0.02$ GiB), demonstrating robust, and computationally efficient performance suitable for real-time deployment after appropriate optimization.

**Lane-wise Performance**   Table 3 presents lane-wise performance of LEO. In the ego lane, the model achieves the highest accuracy, with GIoU above $0.9$ for $l_1$ and $0.84$ for $l_2$, and ($10$–$27$ cm) CP errors, corresponding to the lead vehicle directly ahead of the ego car. This is attributed to favorable sensor coverage and consistent rear-edge visibility of lead vehicles, enabling precise learning of dimensions and orientation. In adjacent lanes, performance degrades moderately (GIoU $0.77$–$0.79$), as sensor placement, FOV, and resolution cause different object edges to be visible for different sensors with varying covariances of extension points for each track. The adaptive fusion mechanism compensates for these differences by weighting inputs through graph attention, yielding robust estimates. Notably, $l_2$ show larger farthest-point errors ($2$–$3$ m), yet the overall high GIoU across lanes substantiates the effectiveness of the proposed approach in handling heterogeneous observability while prioritizing safety-critical objects in the ego lane.

Table 3: Lane-wise analysis for LEO. Values represent mean IoU' (–) and MAE for points.

| Lane ($l_1$ / $l_2$) | GIoU | $CP_x$ | $CP_y$ | $FP_x$ | $FP_y$ |
|---|---|---|---|---|---|
| Ego Lane (EL) | 0.91 / 0.84 | 0.10 / 0.27 | 0.07 / 0.16 | 0.21 / 0.87 | 0.10 / 0.37 |
| Left Lane (LL) | 0.79 / 0.77 | 0.19 / 0.34 | 0.20 / 0.25 | 0.64 / 2.30 | 0.23 / 0.40 |
| Right Lane (RL) | 0.77 / 0.71 | 0.23 / 0.55 | 0.10 / 0.13 | 0.82 / 3.17 | 0.17 / 0.31 |

## 5.2 ABLATION STUDY

**Ablation Study**   LEO's dual-attention mechanism, intra-modal attention for temporal consistency and inter-modal attention $\alpha_{ij}^{\text{inter}}$ for cross-sensor spatial fusion provides a structured interpretation of the degradation patterns. Removing LRR (Table 4) yields the most severe collapse, particularly in the ego lane where only the rear edge of the lead vehicle is typically visible. Radar's longitudinal

Table 4: **Ablation** : Lane-wise analysis for LEO without LRR

| Lane ($l_1$ / $l_2$) | GIoU | $CP_x$ | $CP_y$ | $FP_x$ | $FP_y$ |
|---|---|---|---|---|---|
| Ego Lane (EL) | **0.55** / **0.22** | 1.39 / 0.36 | 0.28 / 0.16 | 2.37 / 13.39 | 0.35 / 0.49 |
| Left Lane (LL) | 0.76 / **0.48** | 0.48 / 0.43 | 0.22 / 1.30 | 1.02 / **7.54** | 0.28 / 0.58 |
| Right Lane (RL) | 0.75 / **0.47** | 0.52 / 0.64 | 0.11 / 0.12 | 1.11 / **7.21** | 0.16 / 0.34 |

Table 5: **Ablation** : Lane-wise analysis for LEO without LRL

| Lane ($l_1$ / $l_2$) | GIoU | $CP_x$ | $CP_y$ | $FP_x$ | $FP_y$ |
|---|---|---|---|---|---|
| Ego Lane (EL) | 0.89 / 0.83 | 0.33 / 0.35 | 0.10 / 0.18 | 0.45 / 1.12 | 0.13 / 0.43 |
| Left Lane (LL) | 0.76 / 0.73 | **0.69** / 0.39 | **0.30** / 0.31 | **1.27** / 2.64 | 0.37 / 0.46 |
| Right Lane (RL) | 0.74 / 0.64 | **0.93** / 0.82 | **0.17** / 0.17 | **1.60** / 3.53 | 0.23 / 0.35 |

penetrability and returns from within the vehicle body supply depth cues that LiDAR and SMPC cannot recover under occlusion; without these signals, intra-modal attention loses its primary constraint on object extent, causing GIoU to drop to $0.55/0.22$ and $FP_x$ to explode to $13.39$ m. LRL ablation (Table 5) produces a different failure mode: its long-range precision and visibility into adjacent lanes are critical for inter-modal spatial aggregation, and removing it markedly increases $CP_x$ and $FP_x$ (up to $1.60$ m), especially under cross-lane occlusions. Thus, removing the LRL destabilizes the stability of the reference point crucial for object tracking and geometric extent.

SMPC ablation (Table 6) produces the mildest degradation: the stereo module primarily contributes near-range depth cues and contour precision, so its removal leads to slight increases in reference-point and lateral errors while largely preserving global object geometry. In contrast, disabling inter-modal attention (Table 2) highlights its essential role in maintaining global multi-sensor consistency. Without cross-sensor relational weighting, reference-point drift increases sharply ($RF_x$: $0.21 \rightarrow 3.98$ m) and dimensional accuracy deteriorates substantially ($l$: $0.43 \rightarrow 3.55$ m, $+83\%$), even though temporal attention remains active. These results underscore the benefit of jointly modelling spatial, temporal, and uncertainty-aware cues within the attention mechanism. Overall, LRR is indispensable for stable longitudinal extent inference under occlusion especially for articulated vehicles. LRL supports cross-lane robustness, reference-point stability, and spatial completeness, whereas SMPC serves as a geometric refinement layer. Finally, inter-modal attention remains fundamental for ensuring globally coherent and uncertainty-consistent multi-sensor shape estimation.

## 5.3 QUALITATIVE ANALYSIS

Figure 6 illustrates a qualitative evaluation of LEO across highway and proving ground scenarios. Learned shapes (magenta) are compared with model-based fusion outputs (green), while sensor tracks from individual modalities are shown in additional colors with velocity vectors as arrows. In highway driving (Figure 6a), input tracks often exhibit shortened bounding box lengths under sparse observations, particularly for distant vehicles. LEO adapts to these degraded inputs while maintaining consistent geometry, and suppresses spurious SMPC detections that erroneously merge multiple objects into one through attention weighting. For articulated objects such as a truck–trailer in the right lane, the model accurately reconstructs the full extent by combining LiDAR contours with near-range SMPC depth cues, outperforming rule-based fusion which systematically underestimates length. In unoccluded cases (Figure 6b), orientation and dimensions align closely with sensor inputs. During dynamic maneuvers such as cut-ins (articulated vehicle merging into ego lane) and emergency braking (Figure 6c), the model produces temporally stable predictions by integrating

Table 6: **Ablation** : Lane-wise analysis for LEO without SMPC

| Lane ($l_1$ / $l_2$) | GIoU | $CP_x$ | $CP_y$ | $FP_x$ | $FP_y$ |
|---|---|---|---|---|---|
| Ego Lane (EL) | 0.89 / 0.83 | 0.25 / 0.33 | 0.09 / 0.21 | 0.36 / 1.04 | 0.12 / 0.51 |
| Left Lane (LL) | 0.75 / 0.71 | 0.43 / **0.45** | 0.26 / 0.28 | 0.95 / 2.91 | 0.31 / 0.49 |
| Right Lane (RL) | 0.75 / 0.66 | 0.64 / 0.70 | 0.15 / 0.18 | 1.24 / 3.46 | 0.22 / **0.39** |

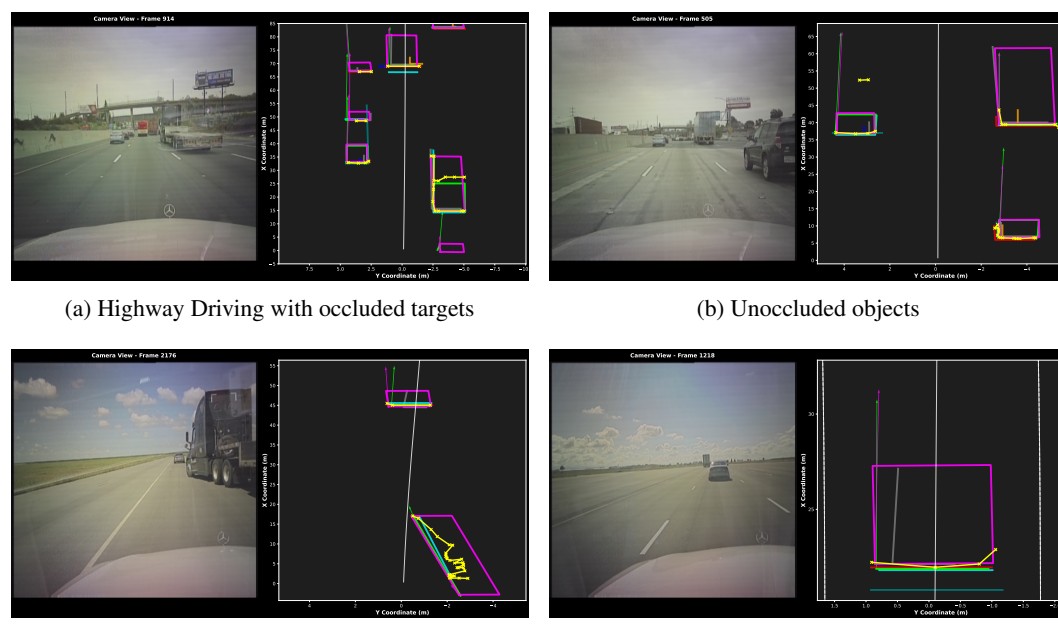

(a) Highway Driving with occluded targets

(b) Unoccluded objects

(c) Articulated Vehicle entering into ego lane

(d) Lead vehicle in ego lane

Figure 6: Evaluation results across diverse real world scenarios.

long-range RADAR length cues with multi-modal velocity estimates, which is critical for safe planning. For near-field targets (Figure 6d), depth inconsistencies across modalities are resolved by prioritizing high-confidence LiDAR contours, yielding corrected and reliable shape estimates.

## 6 CONCLUSION AND FUTURE WORK

This work presented **LEO**, a spatio-temporal GAT-based framework for adaptive shape estimation in extended object tracking, designed under production-level automated driving requirements on track-level sensor inputs. Building on the proposed parallelogram-based ground truth, LEO effectively models both rectangular and articulated target-combination-geometries, while the dual-attention mechanism enables joint reasoning over intra-modal temporal dynamics and inter-modal spatial dependencies for robust multi-sensor fusion. Extensive evaluation on large-scale real-world datasets confirmed that LEO delivers accurate, stable, and computationally efficient shape representations across diverse driving scenarios, validating its suitability for practical deployment. Future research will extend this framework toward uncertainty-aware estimation, domain adaptation, continual learning, and lightweight variants for embedded platforms, as well as integration with planning and decision-making modules to quantify the impact of improved shape-aware perception on automated driving safety and efficiency.

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

# A   APPENDIX

**Sensor Modalities and Delivered Representations.** For clarity, we explicitly list the sensor set and the corresponding output representations available in the Mercedes-Benz DRIVE PILOT dataset. These outputs reflect production-grade perception interfaces and constitute the exact input space used for training and evaluating *LEO*. Unlike raw-sensor academic datasets, the DRIVE PILOT system provides object-level or contour-level entities that encode geometric, kinematic, and uncertainty information.

- **Long-Range Radar (LRR):** Provides tracked object hypotheses with varying geometric detail, including:
    1. Tracked-Point-Objects (High-confidence object hypotheses with estimated position, velocity, and state covariance)
    2. Tracked-Shape-Objects (Extension estimates representing coarse object geometry derived from radar resolution cells)
- **Stereo Camera (SMPC):** Supplies dense geometric and object-level representations, including:
    1. Stixel Cloud (Cordts et al., 2017)(Vertically-oriented depth segments encoding dense geometric structure in regions of interest)
    2. Tracked-Shape (Object-level shape segments generated by the stereo-based tracking module)
- **Long-range LiDAR (LRL):** Delivers high-resolution geometric cues used in the fusion graph, including:

1. LiDAR-Point-Contours (Preprocessed contour segments extracted via geometric feature extraction (e.g., dual-line RANSAC))

2. Tracked-Shape (High-resolution extension estimates derived from LiDAR clustering and tracking)

- **Multi-Mode Radar (MMR):** Produces heterogeneous detection and partial-shape cues, including:

1. Tracked-Points (Sparse point-level detections with Doppler and covariance attributes)

2. Tracked-Partial-Shapes (Partial object edge or segment hypotheses generated when sufficient angular coverage is available)

These heterogeneous, asynchronous sensor deliverables form the nodes and attributes of the spatio-temporal graph used in *LEO*. Their diversity in resolution, update rates, and uncertainty modeling is a key motivation for learning-based fusion. This also explains why existing academic benchmarks (e.g., KITTI, nuScenes, Waymo) are not directly compatible with our input modality or sensor configuration.

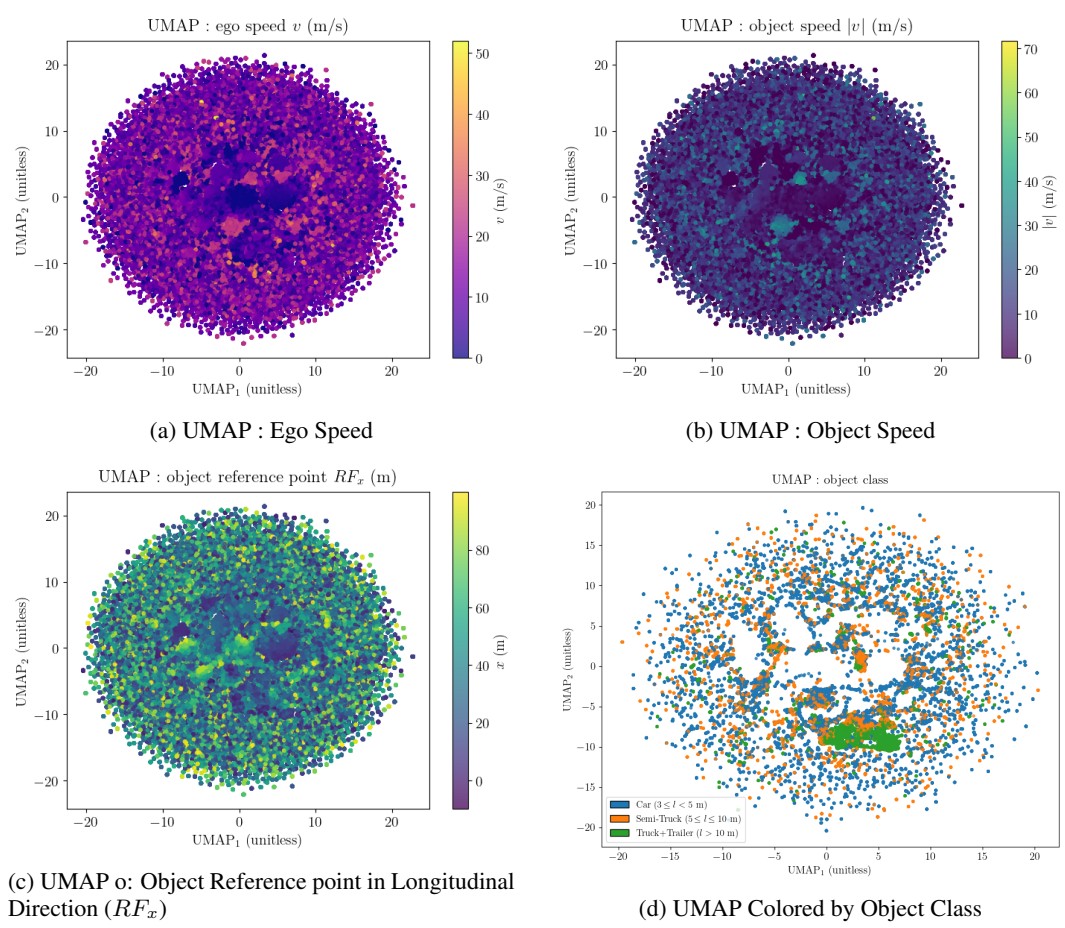

(a) UMAP : Ego Speed

(b) UMAP : Object Speed

(c) UMAP o: Object Reference point in Longitudinal Direction ($RF_x$)

(d) UMAP Colored by Object Class

Figure 7: UMAP embeddings visualized across different object- and ego-level attributes.

The UMAP projections in Fig. 7 highlight the richness and diversity of the DRIVE PILOT dataset. Across all four embeddings, the data form well-structured circular manifolds that smoothly vary with ego speed, target speed, longitudinal object distance, and object class, indicating consistent coverage of the full operational design domain. Such continuity across heterogeneous attributes reflects balanced sampling of driving scenarios and dense annotation quality. Notably, this level of granularity - spanning multi-lane perception, high-resolution extension cues, and a complementary long-range sensor suite - is unique to DRIVE PILOT; no existing open-source dataset provides

comparable longitudinal range, sensor diversity, or fine-scale geometric observability. As a result, the dataset offers an exceptional foundation for learning robust real-world object extent models.

