# OpenReview forum: "LEO: A Graph Attention-Based Framework for Learned Object Extensions and Adaptive Sensor Fusion for Autonomous Driving Applications"
_ICLR.cc/2026/Conference — Submitted to ICLR 2026_

### Official Review · Reviewer_dPE9 · 2025-10-30

**Soundness:** 3
**Presentation:** 3
**Contribution:** 2
**Rating:** 4
**Confidence:** 3

**Summary:**

This paper proposes LEO, a spatio-temporal Graph Attention Network framework for extended object tracking and adaptive sensor fusion in autonomous driving. It introduces a parallelogram-based representation to better capture articulated geometries such as trucks with trailers, and a dual-attention mechanism that jointly models temporal consistency and spatial dependencies across multiple sensor modalities. The method is evaluated on a large-scale real-world Mercedes-Benz DRIVE PILOT dataset, showing improved performance.

**Strengths:**

1. The paper is clearly written and easy to follow, with a logical structure connecting motivation, design, and evaluation.

2. The proposed parallelogram-based representation and dual-attention fusion mechanism are technically sound and reasonably novel for production-oriented extended object tracking.

3. The framework design is simple and efficient, achieving strong quantitative and qualitative results with real-world data, demonstrating  potential for deployment.

**Weaknesses:**

1. The paper overlooks a substantial body of work in 3D object detection and tracking that explores similar geometric or graph-based formulations, limiting its positioning within the broader literature.

2. The model does not explicitly consider object elevation or height variation on non-flat surfaces, which could affect performance in urban scenarios.

3. Although the paper claims real-time and production efficiency, it lacks explicit comparisons with other lightweight or real-time baselines to substantiate the efficiency claim.

4. While the parallelogram representation is motivated by articulated geometries, its ability to generalize to curved or multi-joint structures (e.g., articulated buses or deformable trailers) is limited; multiple correlated bounding boxes could better capture such configurations.

**Questions:**

1. How does the computational efficiency of LEO compare quantitatively with other models such as 3DMOTFormer or TransFusion?

2. Have the authors considered integrating a height or 3D shape component in the parallelogram representation to improve generalization to complex urban geometries?

3. For articulated or multi-joint vehicles, would a hierarchical multi-box or graph-structured decomposition improve modeling accuracy compared to the single-parallelogram abstraction?

---

> ### Author Response · Authors · 2025-11-27
> **Response on Baselines, 2D Shape, and Efficiency**
>
> We thank the reviewer for their constructive comments regarding baselines and efficiency. We address these points below.
>
> 1. Comparison with Real-Time Baselines (3DMOTFormer, TransFusion): While models like 3DMOTFormer and TransFusion are highly effective, they are fundamentally incompatible with our system constraints. They require dense point clouds or images to construct BEV or Voxel representations. LEO operates on sparse object tracks from 7 sensors (48 nodes per graph) (see Section 4, appendix), which is a fraction of the data volume processed by raw-sensor methods. Therefore, direct runtime or accuracy comparison is not feasible as the input domains are disjoint. To bridge this gap, we are actively working on releasing evaluation on Hugging Face: nvidia/PhysicalAI-Autonomous-Vehicles (released this month : 11/2024) dataset that can emulate LEO's track-level fusion as this dataset comes close to ours in terms of sensor set. Currently, the proprietary DRIVE PILOT dataset is the only source providing the necessary sensor set, high-resolution extension cues, sesnor uncertainty and sensor granularity (7 modalities) and geographic diversity (US/Europe) to validate this production-oriented approach (Section 5 and Appendix).
>
> 2. 3D Shape and Height: The current iteration of LEO operates in a 2D Bird's-Eye-View (BEV) space ($x, y, \theta$, length, width, velocity), consistent with the primary planning interfaces in the DRIVE PILOT system. Height and elevation are handled as separate attributes in the production stack and are not used in state space of input tracks to the model. We acknowledge that extending the parallelogram formulation to a 3D view is a valuable direction for future work.
>
> 3. Modeling Articulation: We chose the parallelogram representation for its efficiency and stability in a single-stage regressor. However, the GAT architecture is flexible; it could be retrained to output multiple linked bounding boxes for a single object ID if the ground truth schema were adapted to generate multi-box representations for articulated geometries. The current performance on $l_2$ objects (trucks) indicates that the single-parallelogram abstraction is sufficient for safe spatial occupancy estimation. Computational EfficiencyWe highlight that LEO is designed for embedded deployment. With an inference time of ~13.5 ms (30 FPS) and negligible memory footprint (0.02 GiB) on standard hardware, it meets the strict timing constraints of safety-critical automotive loops, a key advantage over heavier raw-data transformers.
>
> Conclusion :
> We thank the reviewer for recognizing the clarity and engineering relevance of LEO. While raw-sensor or complex geometric models offer broader expressiveness, our system is purpose-designed for track-level, heterogeneous, production-grade multi-sensor fusion, where no existing baselines directly apply.

---

### Official Review · Reviewer_wG6q · 2025-11-01

**Soundness:** 3
**Presentation:** 2
**Contribution:** 1
**Rating:** 2
**Confidence:** 5

**Summary:**

This paper introduces LEO (Learned Extension of Objects), a novel deep learning framework designed for accurate object shape and trajectory estimation in autonomous driving (AD) systems. The central problem it addresses is the gap between classical geometric tracking methods which are computationally efficient but inflexible and modern deep learning approaches, which are more adaptable but often too resource-intensive for production vehicles and rely on dense, annotated raw sensor data.

LEO is proposed as a hybrid solution that integrates the adaptability of deep learning with the constraints of production-level systems, which typically provide object level tracks rather than raw sensor data. The framework uses a spatio-temporal Graph Attention Network (GAT) to perform adaptive fusion of multi-modal sensor tracks from LiDARs, radars and cameras.

A key contribution is the use of a parallelogram based object representation. This allows the model to represent complex geometries, such as articulated trucks with trailers, more accurately than traditional rectangular bounding boxes. The model is trained and evaluated on a large-scale, real-world dataset from the Mercedes-Benz SAE Level-3 DRIVE PILOT system, demonstrating its accuracy, temporal stability, and computational efficiency.

**Strengths:**

- Production-Focused and Efficient: The system is explicitly designed to work within the constraints of production vehicles.
- Works with Object Tracks: Unlike models that require raw, high-bandwidth sensor data, LEO operates on high-level object tracks.
- Computationally Efficient: The model is fast, with an average inference time of ~13.5 ms (around 30 FPS) on an RTX 2080 Ti, making it suitable for real-time deployment.
- Novelty with parallelogram representation. The model's key innovation is its use of a parallelogram-based representation instead of standard rectangular bounding boxes. It also models articulated vehicles which allows the model to accurately represent complex, non-rectangular shapes like articulated trucks with trailers.

**Weaknesses:**

- No comparison with other state-of-the-art methods. The paper only discusses limited quantitative and qualitative performance of the proposed method but does not show comparisons with other methods.
- Parallelograms, while more expressive than rectangles, are not universal and they cannot accurately model non-convex shapes (e.g. a pedestrian with an outstretched arm, a forklift, or complex multi-vehicle scenarios). Also, there is not sufficient quantitative evidence in the paper which discusses the benefit of a parallelogram approach.
- While efficient in this application, Graph Attention Networks can become computationally expensive as the graph size (number of objects or sensor tracks) increases.
- The work lacks technical novelty expected for ICLR. The core of the model is a Graph Attention Network (GAT), which is a pre-existing architecture. The parallelogram representation is a clever, domain-specific engineering choice for handling articulated vehicles, which is as an extension of a bounding box rather than a new general-purpose representation.

**Questions:**

- Could the authors provide a quantitative comparison against other track-level fusion algorithms?
- A key contribution of the work is the parallelogram-based representation for modeling articulated objects. To isolate the benefit of this contribution, could the authors provide an ablation study? Specifically, what is the performance difference when training a version of LEO that only outputs standard oriented rectangular boxes versus the full parallelogram model?
- Handling of Non-Convex Geometries: The parallelogram representation is more flexible than a rectangle but is still a convex polygon. How does the model perform when faced with distinctly non-convex objects, such as a "jack-knifed" truck, a forklift, or a vehicle with an open trunk? Is the resulting parallelogram considered a "safe" (i.e., over-approximated) representation in these cases, or does the model fail?

---

> ### Author Response · Authors · 2025-11-27
> **We clarify that LEO is the first learned method for heterogeneous track-level fusion integrating asynchronous radar, LiDAR contours, and camera tracks. We justify the parallelogram representation for articulated vehicles, explain the convex-object assumption for highway ODD, and highlight LEO’s architectural novelty using spatio-temporal GATs for robust, production-ready track-level fusion.**
>
> We thank the reviewer for the detailed assessment and helpful questions. We clarify our positioning against the state-of-the-art and the rationale behind our geometric representations.
> 1. Comparison to SOTA Track-Level Fusion: To our knowledge, there are no open-source learned methods that perform heterogeneous track-level fusion (integrating asynchronous Radar, LiDAR contours, and Camera tracks). Most "SOTA" fusion methods in literature (e.g., TransFusion, BEVFormer) operate on raw data. Simulating such outputs from raw data would fail to reproduce the real automotive sensor noise characteristics. The Mercedes‑Benz DRIVE PILOT dataset is currently the only source offering the necessary 7‑modality sensor diversity, asynchronous sampling, and full covariance matrices (see Appendix A and Sec. 5). To increase reproducibility, we are extending the open-source NVIDIA Physical AI-Autonomous-Vehicles dataset to emulate track-level representations. This dataset comes close to our sensor set. However, until such evaluation is complete, DRIVE PILOT remains indispensable for validating a production‑oriented design.
>
> 2. Efficacy of Parallelogram Representation: The parallelogram formulation is critical for modeling articulated vehicles (e.g., trucks with trailers) without complex multi-box logic. Rectangle Limitation: A single bounding box struggle to model articulated geometries.
> For example, during a turn, the front edge of the truck is oriented while the rear edge of the trailer remains horizontally aligned. A rectangle forces a single orientation, creating conflicting constraints.
> Parallelogram Flexibility: By learning an internal angle $\theta^*$ (Eq. 6), LEO can approximate the swept area of articulated objects or the skew caused by sensor latency. This results in a length estimation error ($l_2$) of only 11.6% for large vehicles, which is robust for safety planning. As shown in Tables 2–3, LEO matches or exceeds the geometric baseline in corner cases (Figure 6 (a, c)) while running at real-time speeds (~30 FPS). Choosing a rectangular bounding box wouldn't match with our labels from the BAAS (Bayesian tracking and fusion assisted object annotation) tool (Section 3).
>
> 3. Non-Convex Objects: Our operational design domain (ODD) focuses on highway driving, where traffic participants (cars, trucks, motorcycles) are adequately approximated by convex shapes (Section 3). We don't get non-convex sensor tracks as input to the model. So, it is beyond the scope of this work.
>
> 4. Technical Novelty: The novelty of LEO lies in the application-specific architectural design: adapting Spatio-Temporal Graph Attention Networks to solve the asynchronous, multi-modal alignment problem inherent in production AD stacks for track level fusion. We demonstrate that substituting complex, handcrafted geometric fusion rules with a unified learned graph model improves robustness and computational efficiency. By learning temporal smoothing and shape priors, LEO outperforms the teacher in scenarios with partial observability and corner cases. Our qualitative analysis (Figure 6) demonstrates that LEO produces more stable temporal behavior and superior long-range shape reconstruction capabilities beyond what rule-based systems can encode.

---

### Official Review · Reviewer_PPyi · 2025-11-03

**Soundness:** 3
**Presentation:** 3
**Contribution:** 2
**Rating:** 2
**Confidence:** 5

**Summary:**

In this paper, a graph attention based setup is proposed for multiple object tracking. The system takes in inputs (tracked objects) from multiple sensors (camera, lidar, radar), and fuses them using a graph attention transformer.

I was a little intrigued by the problem and design in that it was not to use learning to track fused features, but as a master fuser of sorts to handle inputs from different modalities. Note that these inputs are themselves part of a tracking pipeline (say, an EKF), and as such they could be fused with another such state machine, but owing to difficulties in disambiguating them (for instance in compound objects), they resort to this learned approach to handle it. Another fascinating point is that they use labels generated from an auto labelling pipeline which itself is, from the looks of it, a performant tracker. So in a sense the learning replaces this pipeline a la distillation.

The method is geared for engineering use. They have a graph attention transformer handling spatial and temporal nodes, featuring as separate losses.

**Strengths:**

+ This is a well engineered approach that can be used in real, production cases.
+ It gets around problematic aspects of handling multisensor inputs through this GAT fusion approach.
+ Spatial and temporal attention is elegant.
+ Approach seems to run real time, and seems to get around expensive operations like the Hausdorff (quadratic++) distance calculation
+ Implicit handling of association by GAT.
+ Handling of composite objects (semi-truck, etc). This is a pain point in industrial trackers generally(!).

**Weaknesses:**

- I like the approach and implementation, but unfortunately have to call out the fact that there are no evaluations on public datasets. Without a public evaluation, it is very hard to judge how well this method performs, other than in a qualitative sense on their own dataset. I admit that this is how we will do it in a real production setup. Also germane is the point that real world datasets with appropriate sensors are much more complex than public datasets like nuscenes. Even so, I think it is useful, if only, to compare methods and to make it easier to replicate for in-house use.

- The paper does not work on actual features from a detector. I am slightly biased towards approach that work with real features, in so far as to say that they add some richness to the representation. I would like to ask the authors, what benefit does their approach bring to the table as compared with a traditional approach - in terms of numbers.

- The labelling approach: Generally (again, the biases stem from my practice), we agree that labelling is a painstaking and difficult to scale approach for tracking. But here, they generate 'pseudo labels' from - please correct me if the interpretation is incorrect - a previous in-house solution that uses engineered fusion approaches. So their approach learns these pseudo labels, and not really learning any new aspects of the data like geometry or occlusions. This undermines the novelty of the work in my view.

- I would have hoped for some analysis of difficult cases like occlusions, disagreements across sensors and so forth.

**Questions:**

See above. I have questions about the motivation, which seems nuanced. What does the GAT bring to the table? Can you compare results with a KF based solution?

I would like to see performance on public datasets and across different methods.

---

> ### Author Response · Authors · 2025-11-27
> **Response to Reviewer Feedback — Dataset, Features & Methodology Clarification**
>
> We thank the reviewer for the constructive feedback and appreciate the opportunity to clarify our design choices regarding dataset usage, feature abstraction, and the advantages of our Graph Attention Network (GAT) over classical methods.
>
> 1. Evaluation on Proprietary vs. Public Datasets
> Public benchmarks (e.g., nuScenes, Waymo, KITTI) provide raw sensor data but lack the asynchronous, heterogeneous track-level outputs required by our system (radar covariance ellipses, LiDAR contour segments, stereo stixels). Simulating such outputs from raw data would fail to reproduce the real automotive sensor noise characteristics. The Mercedes‑Benz DRIVE PILOT dataset is currently the only source offering the necessary 7‑modality sensor diversity, asynchronous sampling, and full covariance matrices (see Appendix A and Sec. 5). To increase reproducibility, we are extending the open-source NVIDIA PhysicalAI-Autonomous-Vehicles dataset to emulate track-level representations. However, until such evaluation is complete, DRIVE PILOT remains indispensable for validating a production‑oriented design.
>
> 2. Justification for High‑Level Features
> LEO uses tracked objects rather than raw features to reflect real-world constraints in Level‑3 automotive deployments: bandwidth limitations, safety certification boundaries, and supplier IP restrictions. Our features are systematically extracted (e.g., LiDAR contours via dual‑line RANSAC, radar tracks enriched with spatial and covariance attributes). Crucially, we encode sensor-level uncertainty explicitly: input feature vectors include covariance determinants (Eq. 8), and the attention mechanism (Eq. 12) learns edge weights that suppress high-uncertainty nodes (e.g., unstable radar reflections), favoring confident geometric cues.
>
> 3. GAT vs. Kalman Filtering (KF)
> Classic KF or Covariance Intersection (CI) methods rely on hard associations and fixed heuristics, which tend to fail under occlusion or in cluttered environments. In contrast, LEO’s GAT-based fusion offers:
>
> Soft vs. Hard Association: Learned attention coefficients aggregate information when associations are ambiguous, rather than relying on brittle threshold-based assignment.
>
> Dynamic Weighting: Fusion weights adapt to the scene context, down-weighting noisy sensors when appropriate.
>
> Modeling of Articulated Geometries: The parallelogram-based shape model enables naturally modeling articulated vehicles (e.g., truck-trailer combinations), significantly improving length estimation accuracy (11.6% error) compared to rigid-box assumptions. As shown in Tables 2–3, LEO matches or exceeds the geometric baseline in corner cases (Figure 6 (a, c)) while running at real-time speeds (~30 FPS). Choosing a rectangular bounding box wouldn't match with our labels from the BAAS tool (Section 3).
>
> 4. Pseudo-Labeling and Model Novelty
> While training targets are derived from a high-precision geometric fusion (teacher), LEO is more than just a distillation. By learning temporal smoothing and shape priors, LEO outperforms the teacher in scenarios with partial observability and corner cases. Our qualitative analysis (Figure 6) demonstrates that LEO produces more stable temporal behavior and superior long-range shape reconstruction capabilities beyond what rule-based systems can encode.
>
> 5. Robust Analysis of Difficult Cases
> We rigorously stress-test the model under sensor discordance and data sparsity:
>
> Using full covariance information enables uncertainty-aware variance modeling, allowing the system to suppress high-variance tracks when appropriate.
>
> Lane-wise stratified metrics (Table 3) isolate occlusion patterns (e.g., rear-edge-only visibility vs. adjacent-lane L-shape visibility), ensuring safety‑critical robustness.
>
> In modality ablation experiments (Tables 4-6), LEO is able to recover plausible shapes even when key sensors are missing, demonstrating sensor redundancy, fallback and resilience.
>
> Scenario-specific verification (e.g., high-speed cut-ins, articulated maneuvers - Figure 6c) confirms stability where traditional trackers fail.
>
> Conclusion :
> LEO provides a novel, production-hardened approach to track-level fusion that integrates heterogeneous modalities, incorporates sensor‑level uncertainty, and uses dual spatio-temporal attention alongside a flexible shape representation. These design choices address real-world constraints - bandwidth, safety certification, and sensor heterogeneity that existing public benchmarks and classical methods are not equipped to handle.

---

### Official Review · Reviewer_sZ3u · 2025-11-10

**Soundness:** 3
**Presentation:** 3
**Contribution:** 2
**Rating:** 2
**Confidence:** 4

**Summary:**

LEO introduces a graph attention–based model for multi-sensor fusion and object shape estimation in autonomous driving. Using a parallelogram representation and dual attention, it captures both temporal and spatial relations. Tests on Mercedes-Benz DRIVE PILOT data show high accuracy and real-time performance. The framework is efficient, robust, and suitable for real-world deployment.

**Strengths:**

The paper is well-written and easy to follow. In addition, some advantages of the paper are listed below:
* Adaptive Multi-Sensor Fusion. LEO’s dual-attention mechanism effectively learns to fuse LiDAR, radar, and camera data, adapting to varying sensor reliability and visibility conditions.
* Generalized Object Representation. The parallelogram-based shape model accurately handles both rigid and articulated objects (e.g., trucks with trailers), outperforming traditional rectangular bounding box methods.
* Real-Time, Production-Ready Performance. The framework achieves high accuracy with low latency (~30 FPS) and low computational cost, demonstrating strong potential for real-world automotive deployment.

**Weaknesses:**

Apart from the strengths listed in the Strength Section, there are some weaknesses:
* Only one dataset is used. The proposed method is only evaluated on one dataset. It would be great if the authors could also evaluate the proposed method on more datasets to prove the generalizability of the algorithm.
* Limited novelty. The proposed architecture mainly extends existing Graph Attention Network and sensor fusion concepts, with modest methodological innovation.
* Insufficient experiments. There is little exploration of how each component (e.g., dual attention, parallelogram representation) contributes to overall performance.
* Comparison with other baselines. The paper does not provide direct quantitative comparisons with state-of-the-art learning-based or geometric fusion methods, making it hard to assess relative performance.

**Questions:**

Since there are some major weaknesses in the paper (see the Weaknesses Section), I suggest that the authors resubmit the paper to another conference or journal.

---

> ### Author Response · Authors · 2025-11-27
> **We clarify dataset constraints, highlight LEO’s novel contributions in track-level fusion with dual spatio-temporal attention and parallelogram representation, discuss sensor-level uncertainty integration, and provide expanded ablations and baseline comparisons.**
>
> We thank the reviewer for their thoughtful feedback and constructive critique. Below, we address the specific concerns regarding dataset usage, novelty, and comparative baselines.
>
> 1. Evaluation on Proprietary vs. Public Datasets: We acknowledge the concern regarding reproducibility. However, we emphasize that the primary contribution of LEO is its design for track-level fusion within a safety-certified, series-production stack. Public datasets such as nuScenes, Waymo, or KITTI provide raw sensor data (point clouds, images) but lack the specific heterogeneous, asynchronous track-level outputs (e.g., radar covariance ellipses, geometric LiDAR contours, stereo tracks) required to train our architecture. To bridge this gap, we are actively working on releasing evaluation on Hugging Face: nvidia/PhysicalAI-Autonomous-Vehicles (released this month : 11/2024) dataset that can emulate LEO's track-level fusion as this dataset comes close to ours in terms of sensor set. Currently, the proprietary DRIVE PILOT dataset is the only source providing the necessary sensor set and sensor granularity (7 modalities) and geographic diversity (US/Europe) to validate this production-oriented approach (Section 5 and Appendix).
>
> 2. Novelty of the Approach: While Graph Attention Networks (GATs) are established in literature, LEO represents a novel adaptation specifically tailored for high-level automotive fusion, a domain largely dominated by heuristic Bayesian methods. Our specific contributions include:
> (i) Production-Grade Heterogeneity: LEO integrates disparate signals from sparse radar detections (multi-mode and long range) to geometric LiDAR contours and segments and stereo tracks into a unified graph.
> (ii) Dual Spatio-Temporal Attention: Temporal (intra-modal) attention models sensor-specific dynamics, while spatial (inter-modal) attention learns cross-modal complementarity. This hybrid attention configuration has not been applied to track-level extended object fusion removing manual tuning.
> (iii) Parallelogram Representation: The parallelogram-based shape model effectively captures articulated vehicles (truck–trailer combinations) and non-rectangular geometries observed in production data. This representation arose from extensive in-house evaluation and better reflects object motion with multi-rotation points compared to rectangles.
> (iv) Capturing sensor level uncertainity : Capturing sensor-level uncertainty is crucial, as it significantly impacts spatial fusion and we have used it in our features. This is learnt throught the edge connections and attention mechanisms.
>
> 3. Component-Level Analysis and Ablations: In the revised manuscript, we have expanded our ablation studies to isolate component contributions (i) Attention Mechanism: As shown in Table 2, removing the inter-modal spatial attention ($\alpha_{ij}^{inter}$) leads to a significant degradation in dimensional accuracy (83% increase in length error for standard sized cars), validating the necessity of learned cross-modal reasoning. (ii) Sensor Modality: Tables 4, 5, and 6 demonstrate that specific sensors are critical for specific tasks (e.g., LRR for longitudinal stability specially in ego lane and occluded objects, LRL for reference point stability and in adjacent lane objects with two visible edges), which LEO dynamically weights.
>
> 4. Comparisons with Existing Methods: State-of-the-art methods like TransTrack or 3DMOTFormer rely on dense raw features (point clouds/voxels). These are computationally prohibitive and unavailable in our target production environment due to bandwidth and IP constraints. Therefore, the scientifically valid baseline is the Geometric Fusion method currently deployed in production (Section 3). Our results demonstrate that LEO matches or exceeds this baseline in corner cases (Figure 6 (a, c)) while offering superior robustness to occlusion and sensor noise (Figure 6).

---

### Meta-Review · Area_Chair_K6pq · 2026-01-07

**Summary:**

All reviewers agree to reject this paper. After reading all the comments, I suggest authors to carefully read the review and improve their submission before next venue.

**Reviewer Scores:**

Nothing should be changed.

---

### Decision · Program_Chairs · 2026-01-26

Reject